# Relationships between Informal Sports Leadership and Emotional Intelligence: A Cross-Sectional Study

Isabel Mercader-Rubio [1] , Nieves Gutiérrez Ángel [1,*] , Ana Teixeira [2,3,4] and Sónia Brito-Costa [3,4,5,*]

1   Department of Psychology, Faculty of Education Sciences, Universidad de Almería, La Cañada, 04120 Almería, Spain; imercade@ual.es
2   Institute of Electronic Engineering and Telecommunications of Aveiro (IEETA, UA), 3810-193 Aveiro, Portugal; anateixeira@esec.pt
3   Research Group in Social and Human Sciences (NICSH), Coimbra Education School, Polytechnic of Coimbra, 3045-043 Coimbra, Portugal
4   Polytechnic Institute of Porto, Center for Research and Innovation in Education (InED), School of Education, 4200-465 Porto, Portugal
5   Institute of Applied Research (i2A), Polytechnic of Coimbra, 3045-043 Coimbra, Portugal
*   Correspondence: nga212@ual.es (N.G.Á.); sonya.b.costa@gmail.com (S.B.-C.)

**Abstract:** Sports leadership research has increased in recent years in sport psychology and is now considered an indication of variables such as sport efficacy, satisfaction and happiness with sport, and emotional well-being. The aim of this manuscript is to analyze the relationship between informal sports leadership and each of the dimensions of emotional intelligence. Two validated and standardized instruments were applied to measure each of the psychological constructs, respectively, that make up the thematic axis of this research. Thus, to measure self-perceived leadership, we used the Sport Leadership Behavioral Scale and its Spanish version, the Sport Leadership Behavioral Inventory. On the other hand, to measure the self-perceived emotional intelligence of the participants, we used the Trait Meta Mood Scale (TMMS-24). The total number of participants was 163 students with a mean age of 20.33 years (SD = 3.44) from university degrees related to physical activity and sport sciences. The main contribution of our research lies in the demonstration that there is a statistically significant relationship between clarity and emotional regulation with empathy (=0.18, $p < 0.001$), decision making (=0.08, $p < 0.001$), social support l (=0.19, $p < 0.001$) and sports values (=0.01, $p < 0.001$). This indicates that the definition of a leader is a person who is aware of their own emotions and those of their group, who fosters positive, communicative, supportive inter-relationships based on sporting values, through the exercise of a positive influence on the other members of the group. In conclusion, this research demonstrates the existence of a significant connection between the components of emotional intelligence (attention, clarity and regulation) and sports leadership (social support, empathy, sports values, decision making and task orientation). Thus, this study concludes that this link is characterized by being direct and positive.

**Keywords:** leadership; sport; emotional skills; university students

## 1. Introduction

Our research covers two major psychological constructs: leadership and emotional intelligence. Thus, when we talk about leadership, we are referring to the ability to influence the thoughts, behaviors, and feelings of athletes so that they enthusiastically pursue the achievement of individual and collective objectives beneficial to the group. In this research, we focus on informal leadership which is articulated around: social support (understood as help and encouragement in positive and negative situations by the leader), empathy (understood as the emotional identification of the leader with the team), sports values (the representativeness of the leader in positive intangible qualities, such as honesty, reliability and responsibility), decision making (referring to the group and individual decisions that

the leader must make) and task orientation (refers to instrumental behaviors guided by the achievement of sports objectives and results). Emotional intelligence is understood as the ability to know, understand and regulate both one's own and others' emotions. According to skill models, this is made up of three essential skills, which are attention, clarity and emotional regulation.

## 2. Theoretical Review of the Topic

Sport increases the quality of life of individuals who participate in it, both physically and psychologically, by reducing depression and stress-related ailments. It also helps with mood [1–3], and sport's social and psychological benefits are contingent on organization, confidence, and leadership abilities. In sports psychology, it is regarded as an indication of well-being, efficacy, a favorable environment, and satisfaction with sport [4]. The study of this style of leadership has become a relevant part of the optimization, evolution, and productivity of both the individual player and the team in recent years [5–7].

Leadership is defined as the coach's ability to influence the thoughts, behaviors, and feelings of athletes so that they enthusiastically pursue the achievement of individual and collective objectives beneficial to the group. The coach is the main actor in the interactions that are established between players and coaching staff on daily basis [8]. However, several conceptual approaches in leadership research have been developed in order to better uncover the most appropriate characteristics, competences, attitudes, behaviors and styles that promote the most effective leadership possible [9–11].

When it comes to sports effectiveness, sports leadership is currently one of the most important issues in sports psychology [12]. It should be highlighted, however, that it has a broad terminological framework with numerous definitions [13–15]. Despite these differences, all of these definitions agree that it is a phenomenon characterized by its processual component, in which the impact on other group members leads to the achievement of a common goal [16]. Sports leadership is carried out by someone who has influence over others, is communicative and is conducive to success, and who is well regarded by the rest of the influential group [16,17], given the existence of a group with common objectives and a process of influence between the leader and their followers [18].

Adjusting to the field of leadership in sport, there are several studies that have established relationships between the type of leadership exercised by the coach and the performance of athletes [9,19–21]; the multidimensional model of leadership [22,23] is the most prominent model for our study.

Derived from Chelladurai's model [9], the Leadership Scale for Sport (LSS) was created to evaluate leadership in sport, proposing that team satisfaction depends on the degree to which the leader's conduct consistently brings together the behaviors and skills necessary to achieve the objectives. However, it must be considered that the leader's behaviors are determined by his or her own characteristics, such as professional competencies [24,25].

In other words, one perspective of comprehending this concept arises from the role of the coach, captain, or another team member, which holds true in both professional and amateur sports and in which the presence of two forms of leadership, formal and informal, is also considered. Although there are similarities between both types of leadership, they cannot be analyzed using the same terms because the formal leader responds to a hierarchical and authoritarian command structure (generally developed by the coach) [26,27]. This is where most of the research on leadership in sport has focused [6]. However, in this study, we are going to focus on informal leadership, which is theoretically based on the contributions of role differentiation theory [28,29] and role integration theory [30–36]. This type of leader is defined as someone who favorably influences the rest of his colleagues [37–39], where the idea of influence is not subject to the prior structure of the group [6], and in which there have been fewer studies that have focused on this role [40].

This two-dimensional model [6] is made up of the following variables: social support (understood as help and encouragement in positive and negative situations by the leader); empathy (understood as the emotional identification of the leader with the team); sports

values (the representativeness of the leader in positive intangible qualities, such as honesty, reliability, and responsibility); decision making (referring to the group and individual decisions that the leader must make); and task orientation (refers to instrumental behaviors guided by the achievement of goals and sports results).

Then, we must differentiate the existence of different types of leadership that coexist in sports teams, such as the leadership role played by a player who has a positive influence on the rest of the members to achieve common goals for all of them. Research on informal athlete leadership has received less attention, both theoretically and concerning the instruments used for its assessment [41,42]. This differentiation between formal and informal leaders, mentioned above, is especially relevant in sports teams, where the coach is a formal leader who can coexist with a team player who is an informal leader [33,43,44].

Other aspects of leadership, such as decision making, motivating participants, reporting viewpoints, developing interpersonal ties, and confidently leading the team, are included in the concept of leadership in the sphere of sports [20,45]. Emotional intelligence, on the other hand, is a new psychological construct that has received a lot of study interest [46–49]. It is regarded as a particularly influential psychological capacity in the emotional aspects of athletes [50], but also in other areas such as decision making and performance [51,52].

It is worth mentioning that two theoretical contributions to the identification and comprehension of this idea coexist: mixed models and skill models. In this research, we concentrate on skill models [52], which differentiate themselves by the fact that they are composed of many skills, each of which comprises a series of sub-skills that correspond to [53–56]:

- Emotional perception: this is the capacity to match and examine one's own and other people's feelings.
- Emotional understanding: this involves knowing, organizing, and retrospectively inspecting one's own and others' emotions.
- Emotional regulation: this is about acquiring, investigating, and thinking about emotions to exploit them to the maximum, both interpersonally and intrapersonally.

As a result, the main reason for the treatment and investigation of emotional intelligence in the sports field is the importance of understanding emotions [7,56–58], which is based on sports psychology contributions [59] that consider it a factor that favors performance, well-being, life satisfaction, group cohesion [60–63], and motivation [55,62,64,65]. It is a skill that enables the athlete to successfully tackle the various scenarios he encounters, providing the athlete with the emotional control needed to manage his emotions effectively and efficiently. Based on all of these theoretical contributions, we decided to investigate the relationship between the two exposed psychological constructs: emotional intelligence and leadership, as viewed from a social and task-oriented perspective, which are divided into five basic components: empathy, social support, influence on decision making, sports values, and task orientation.

The research question to be answered is as follows: "can promoting and fostering emotional intelligence in future professionals in physical activity and sports sciences improve students' leadership capacity?" This research topic is the impetus for selecting the sample for our study, which corresponds to students pursuing degrees in physical activity and sports sciences, and thus potential professionals in this field. Specifically, we selected students studying degrees in Physical Activity and Sports Sciences, Primary Education (specializing in Physical Education), Secondary Education Teacher Training (specializing in Physical Education), and Research in Physical Education Physical Activity and Sport.

We were interested in knowing their self-perceptions of emotional intelligence and leadership for two fundamental reasons: On the one hand, such results could provide us with information about the student's knowledge in terms of intelligence, emotional intelligence, and leadership, as well as their needs to address them in greater or lesser depth from subjects such as sports psychology. On the other hand, because the purpose of this study is to examine the relationship between each of the variables that comprise

the two-dimensional model suggested in relation to sports leadership and each of the dimensions of emotional intelligence.

Despite the fact that both psychological constructs have been widely addressed in psychology in general and sports psychology in particular, there is a paucity of published literature on the subject, which is equivalent to a limitation of this research because such a situation makes it difficult to understand the psychological processes involved in informal leadership in sports and how emotional intelligence influences this.

The research question that guides this research refers to knowing what the relationship between informal leadership and emotional intelligence is. Therefore, our main objective is to study the relationship between informal leadership and emotional intelligence.

As for specific objectives, we propose to analyze the influence of each of the dimensions of both psychological constructs.

Regarding social support, our objective is to analyze the relationship between it and attention, emotional clarity and regulation, and social support. For this, the following hypothesis was formulated:

-   There is a statistically significant relationship between attention, emotional clarity and regulation, and social support.

Regarding empathy, our objective is to investigate the relationship between it and attention, emotional clarity and regulation, and social support. For this, the following hypothesis was formulated:

-   There is a statistically significant relationship between attention, clarity and emotional regulation and empathy.

Regarding values, our objective is to explore their influence on attention, emotional clarity and regulation, and social support. For this, the following hypothesis was formulated:

-   There is a statistically significant relationship between attention, clarity and emotional regulation and sports values.

Regarding decision making, our objective is to study the relationship between decision making and attention, emotional clarity and regulation, and social support. For this, the following hypothesis was formulated:

-   There is a statistically significant association between attention, emotional clarity and regulation, and decision making.

Regarding task orientation, our objective is to examine what relationship exists between it and attention, emotional clarity and regulation, and social support. For this, the following hypothesis was formulated:

-   There is a statistically significant relationship between attention, clarity and emotional regulation and task orientation.

This study will allow us to delve into this topic, which has received little research attention, and will allow us to fill the gap that exists regarding the topic. In this way, our research will help improve understanding of both constructs, as well as the development of more elaborate knowledge and effective approaches and strategies to improve sports leadership and promote the emotional well-being of athletes.

## 3. Materials and Methods

### 3.1. Participants

The total number of participants was 163 students, where 70.9% (n = 117) were men and 27.9% (n = 46) were women. The total sample is made up of students of university degrees related to physical activity and sport sciences (both the Bachelor's degree in Physical Activity and Sport Sciences and the official Master's degree in research in Physical Activity and Sport Sciences). The mean age was 20.33 years (SD = 3.44). The sample size was calculated using Soper's models. The reason for choosing students related to degrees related to physical activity and sports sciences is due to our interest in knowing the current situation and the perception of this group with respect to the two psychological

constructs that are the alignment of this research: informational leadership and emotional intelligence. Thus, the information obtained provides feedback to teachers of these degrees on what the students know, master, or require further training in from the field of sports psychology (Table 1).

**Table 1.** Sociodemographic characteristics of the participants: age, sex and course.

| Course | Females | Males | Total | >25 Years | <25 Years |
|---|---|---|---|---|---|
| 1st | 18 (39.1) | 68 (58.1) | 86 | 86 (97.8%) | 2 (2.2%) |
| 2nd | 16(34.8) | 23 (19.7) | 39 | 38 (97.4) | 1 (2.6%) |
| 3rd | 6 (13%) | 14 (12%) | 20 | 20 (100%) | 0 |
| Total | 40 | 105 | 145 | | |
| 1st Master's degree | 6 (13%) | 12 (10.3) | 18 | 4 (22.3%) | 17 (77.7%) |
| Total | 46 | 117 | 163 | | |

### 3.2. Formalities

First, a questionnaire was given to all participants who attended class on a previously agreed day and with the consent of the teacher responsible for the subject. All the participants were informed of the aims of the study, as well as the teachers of the classes in which the survey was carried out. The chosen inclusion criteria were that the students who participated were registered in the matter and course in question, that they accepted their participation across informed consent (official model of the University of Almería) and that they were of legal age. Once the informed consent was obtained, the questionnaire was administered. On the other hand, as exclusion criteria, instruments that were not complete or in which information was missing were discarded. The total sample size is conclusive due to the total number of students who gave their prior informed consent and decided to participate in our research.

The sample size was calculated using the Soper a priori sample size calculator for structural equation models (Soper, 2022 [66]). Thus, based on six observable variables, one latent variable, with an anticipated effect size of 0.30, probability level of 0.05, and desired statistical power level of 0.95, the minimum recommended effect size was 200 cases, meaning the number of participants in our study was close enough to the population size suggested.

### 3.3. Ethics Statement

This research has the approval of the Institutional Review Board of the University of Almería under the reference: UALBIO2022/035.

### 3.4. Measures

We used the Sport Leadership Behavioral Scale and its Spanish counterpart [40] as well as the Sport Leadership Behavioral Inventory [35] to assess self-perceived leadership. This scale consists of 29 items divided into the following categories: social support (6 items), empathy (5 items), sporting values (6 items), decision making (6 items), and task orientation. (6 items) using a Likert scale (1–5). Cronbach's alpha = 0.94 was achieved for the entire scale. Furthermore, it demonstrated good reliability for each subscale: social support (=0.92), empathy (=0.87), sporting values (=0.90), influence on decision making (=0.87), and task orientation (=0.90).

To assess self-perceived emotional intelligence, we employed the Trait Meta Mood Scale (TMMS-24) [67]. It is a scale composed of 24 items split among three factors: emotional attentiveness (items 1–8), emotional clarity (items 9–17), and emotional regulation (items 18–24) using a Likert-type scale (1–5). The scale reflects an adequate internal consistency (Cronbach's alpha = 0.84), which additionally was observed in its three dimensions (perception, =0.90; clarity, =0.90; and regulation, =0.86).

### 3.5. Procedure

We calculated the hierarchical omega [68] and the explained common variance (ECV) in order to know the relationship of a latent variable on a conglomerate of items. Lss results ≥0.70 indicate the existence of a one-dimensional structure [69]. Regarding ECV, results lower than 0.70 correspond to multidimensionality and results higher than 0.85, with unidimensionality. In addition, ECV allowed us to identify the GFR, for which results ≥0.80 are significant with GFR [69]. In summary, results >0.70 indicate the fact that a latent variable is well defined by its indicators [69].

The procedure to carry out the investigation consisted of five steps:

STEP ONE: Request for consent and permission to carry out the research to the Institutional Review Board of the University of Almería.

STEP TWO: Prior to collecting the information, we contacted a professor from each of the courses of the different degrees. After making contact, we established an appointment to go to the classroom. The responsible professor was informed of the duration and objectives of the investigation.

STEP THREE: Once we went to each classroom, the main researcher explained to the students the purpose of the research and guaranteed the confidentiality of their information, and we acquired voluntary participation in the research.

STEP FOUR: Before filling in the questionnaire, the students filled out the official informed consent of the University of Almería (Spain).

STEP FIVE: Due to the totality of the students who offered their consent and decided to participate voluntarily, the total sample that makes up our research can be considered conclusive. Of the total 277 participants of the study, 165 were part of this investigation (See Figure 1).

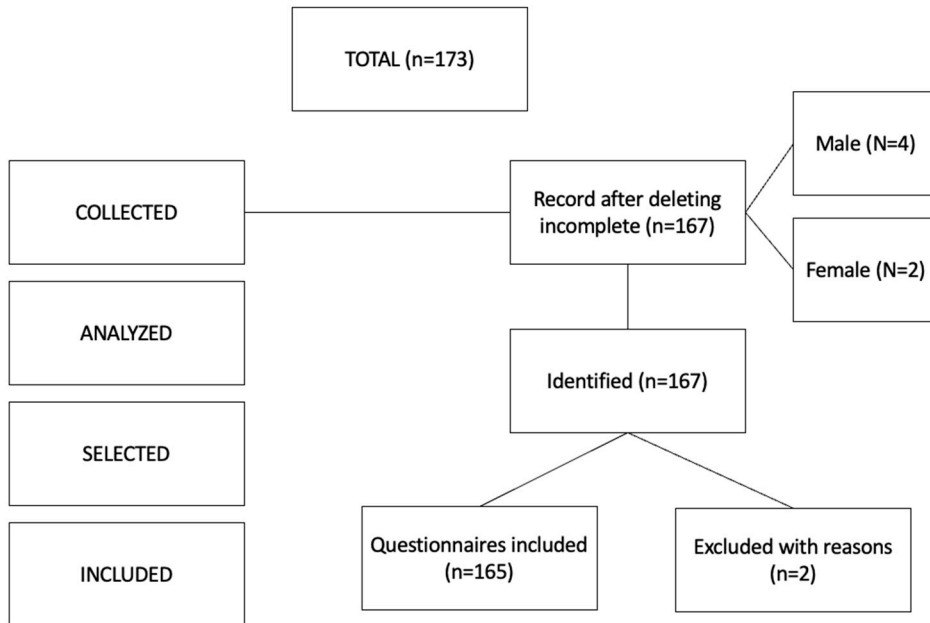

**Figure 1.** Flowchart of the selection procedure.

This research corresponds to an observational and descriptive cross-sectional study, which can be defined as quantitative, based on an exploratory and explanatory analysis based on obtaining data obtained from the administration of different validated and standardized questionnaires that were used for said investigation.

### 3.6. Data Analysis

All the analyses were carried out using the programs: SPSS (version 26) and the R (version 2015). The analysis modules belong to the "lavaan" package. The descriptive

statistics data were calculated in terms of performing different tests such as the mean, standard deviation, bivariate correlations, and internal consistency of each subscale ($\alpha$).

Once these scores were obtained, reliability was analyzed, and structural equation modeling (SEM) was performed to test the relationships established in the hypothetical model. Regarding structural equation modeling (SEM), we also performed the Joreskog test [70,71] and a multiple cause indicator (MIMIC), considering these tests are statistically the most appropriate to inquire about the latent variable (which is made up of a set of measurements).

To consider the model valid or invalid, we took into account the results obtained in the following indices: Tucker–Lewis Index (TLI), standardized mean square residual (SRMR) and mean square error of approximation (RMSEA) [72].

## 4. Results

### 4.1. Results of the Relationship between Emotional Intelligence and Empathy, Decision Making and Task Orientation

The results indicate the following bivariate correlations between each of the dimensions of emotional intelligence (AE/CE/RE) and their relationships with empathy, decision making and task orientation (See Table 2).

**Table 2.** Preliminary analyses.

|  | AE | CE | RE | 1 | 2 | 3 |
|---|---|---|---|---|---|---|
| AE |  | 0.17 | 0.263 ** | 0.177 ** | 0.051 | 0.187 * |
| CE |  |  | 0.508 ** | 0.154 * | 0.064 | 0.245 ** |
| RE |  |  |  | 0.079 | 0.153 * | 0.131 |
| 1 |  |  |  |  | 0.320 ** | 0.276 ** |
| 2 |  |  |  |  |  | 0.494 ** |
| 3 |  |  |  |  |  |  |

EA: emotional attention; EC: emotional clarity; RE: emotional regulation; 1: empathy; 2: decision making; 3: task orientation. Note * $p < 0.05$; ** $p < 0.01$.

Our structural equation model (SEM), (Figure 2), obtained the following scores:

- global fit indices (evaluates the model in general): $p < 0.001$, RMSEA = 0.07, GFI = 0.94.
- Incremental or comparative fit indices (equates the proposed model with the model of independence or absence of relationship between the variables): NNFI = 0.81; TLI = 0.73; IFC = 0.86; IFI = 0.87.
- Parsimony indices (which measure the quality of the fit of the model based on the number of coefficients estimated to reach said level of fit): AGFI = 0.81.

In addition, the scores of two indicators were analyzed and considered: composite reliability (CR), which indicates acceptable reliability [73], and the AVE (average variance extracted) index, which evaluates the variance captured by a construct in relation to the other [74].

The results show the relationships established by the structural equation model and their relationship with empathy, task orientation, and decision making.

Thus, our findings indicate that empathy is positively related to emotional clarity and emotional regulation (=0.18, $p < 0.001$). They are favorably correlated with emotional clarity and emotional management (=0.08, $p < 0.001$).

In other words, individuals who know, organize, and retrospectively evaluate their own and others' emotions, as well as having mastery of investigating and rethinking their own and others' emotions, have greater capacities for empathy and decision making.

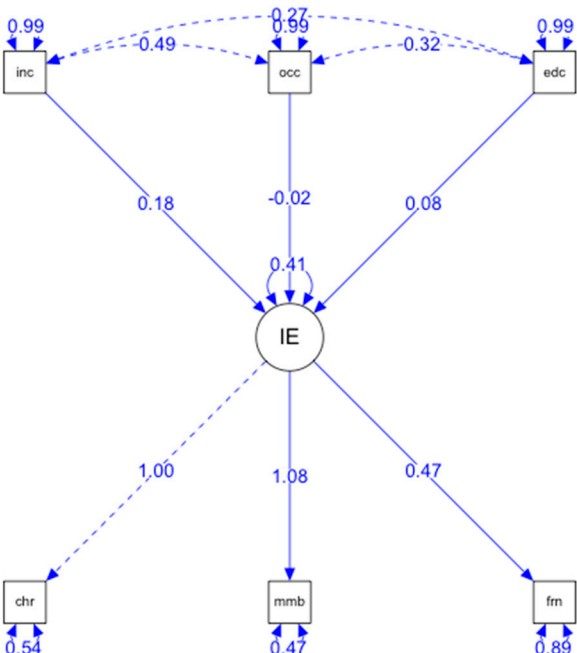

**Figure 2.** Structural equational modeling. Inc: empathy; Occ: or task; Edc: decision making; Chr: ae; Mmb: emotional clarity; Fm: emotional regulation; IE: emotional intelligence.

### 4.2. Results of the Relationship between Emotional Intelligence and Social Support and Sports Values

The relationship between emotional intelligence and decision making and task orientation, on the other hand, shows the following bivariate correlations between each of the emotional intelligence dimensions (AE/CE/RE) (See Table 3).

**Table 3.** Preliminary analyses.

|  | AE | CE | RE | 4 | 5 |
|---|---|---|---|---|---|
| AE |  | 0.17 | 0.263 ** | 0.16 | 0.054 |
| CE |  |  | 0.508 ** | 0.133 | 0.029 |
| RE |  |  |  | 0.161 * | 0.095 |
| 4 |  |  |  |  | 0.786 ** |
| 5 |  |  |  |  |  |

EA: emotional attention; EC: emotional clarity; RE: emotional regulation; 4: decision making; 5: task orientation. Note * $p < 0.05$; ** $p < 0.01$.

For the structural equation model (SEM) (Figure 3), the scores obtained were the following:

- global adjustment indices: $p < 0.001$, RMSEA = 0.08, GFI = 0.96.
- Incremental or comparative fit indices: NNFI = 0.85; TLI = 0.76; IFC = 0.89; IFI = 0.90.
- Parsimony indices: AGFI = 0.85.

The results provided exhibit the correlations developed in the structural equation model, as well as their relationship with social support and sporting ideals. They also show an advantageous connection between emotional clarity and emotional regulation and social support (=0.19, $p$ 0.001). Also, there was a link between emotional clarity and emotional regulation with sporting values (=0.01, $p$ 0.001). However, such correlation with emotional attention is not positive.

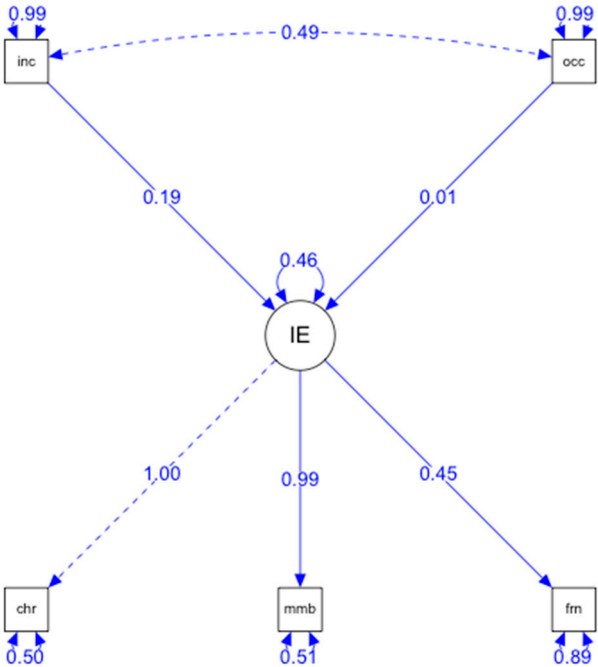

**Figure 3.** Structural equational modelling. Inc: social support; Occ: sporting values; Chr: ae; Mmb: emotional clarity; Fm: emotional regulation; IE: emotional intelligence.

## 5. Discussion

The purpose of this study was to investigate the relationship between each of the dimensions of sports leadership and each of the three subskills related to emotional intelligence. The main contribution of our study is the fact that it tries to analyze informal leadership, which has been much less studied in the scientific literature. In addition to this, our research sheds light on the existing gap in that it also investigates the relationship between informal leadership and emotional intelligence.

Considering each of the proposed hypotheses, all of them are partially fulfilled because our results are conclusive with two dimensions of emotional intelligence (emotional clarity and emotional regulation), but not with emotional attention. Therefore, the main findings of our research correspond to the fact that the importance of emotions in the field of sport has been demonstrated. Specifically in the role of the sports leader, which on an emotional level corresponds to a person who considers his own emotions and those of his group [6] with whom he maintains a type of positive, communicative, supportive relationship based on sports values [41,75]. That is, he belongs to a group in which he exerts a positive influence [76] on the other athletes belonging to the group, also based on social support [33,77].

These results are similar to those obtained by other scientific studies that have aimed to analyze informal sports leadership, in which other psychological variables have been analyzed, such as motivation, anxiety, or basic psychological needs [43,55]. However, none of them have covered the influence of emotional intelligence on this type of leadership, which is a novelty in this research. Without a doubt, leadership is a topic that is gaining more and more importance in sports psychology and has become a widely studied topic. The prevalence and rise of publications focused on the study of emotional intelligence from the discipline of psychology in the 21st century is unquestionable. In fact, the novelty of this study identifies the importance of investigating this psychological construct in the field of sports psychology. This, in turn, also corresponds to an area of knowledge that has increasing relevance, interest, and impact [78–80].

Therefore, the conclusions of this work reveal the relationship between different aspects such as the athlete's performance, the validity of training, social skills, problem solving and leadership (which is so important in the sports field), and emotional intelligence.

Regarding the educational implications of these results, they indicate the importance of instructing future professionals in physical activity and sports sciences, not only in the importance of physical training but also of cognitive training, mainly, emotional. We consider the treatment of the importance of emotions in the sports field to be of great relevance, not only for physical activity and sports professionals, but also for those who are in their training period in programs and specific subjects related to it. However, our study is limited and care must be taken with the generalization of the results because one of the great limitations of this study corresponds to the sample size. One of the possible improvements to the quality of this research lies in replicating it and increasing the sample size, with a double purpose: to verify the results provided by this study, and to delve deeper into the field of conducting a cross-sectional study to verify how.

The passage of time and the growth in training may have an impact on the outcomes given in this paper. Furthermore, we must acknowledge as a restriction the fact that there is little scientific research on the relationship between leadership and emotional intelligence, despite the fact that there are countless publications on both sports leadership and emotional intelligence. Another drawback is that it is a cross-sectional study in which the influence of time on the change or stability of the variables evaluated was not measured.

As well as the analysis of these variables, taking sports teams or representatives of various team disciplines as a research sample would be beneficial.

## 6. Conclusions

This study revealed a significant relationship between emotional intelligence (attention, clarity, and regulation) and sports leadership (social support, empathy, sports ideals, decision making, and task orientation). Therefore, this study demonstrates that this link is distinguished for being direct and positive in two of the three dimensions of emotional intelligence.

Thus, future professionals involved in sports-related professions in these areas must be reminded of the significance of additional research. This is crucial in order to emphasize emotional training over physical or cognitive training promotion.

Additionally, programs aimed at developing, improving, and raising awareness of emotional intelligence within organizations are potentially helpful.

**Author Contributions:** Conceptualization and Methodology: I.M.-R., N.G.Á. and S.B.-C.; software: N.G.Á., A.T. and S.B.-C.; formal Analysis: N.G.Á. and S.B.-C.; investigation: I.M.-R., N.G.Á., A.T. and S.B.-C.; resources: I.M.-R., N.G.Á. and S.B.-C.; data curation: I.M.-R. and N.G.Á.; writing—original draft preparation: N.G.Á., I.M.-R. and S.B.-C.; writing—review and editing: N.G.Á. and S.B.-C. All authors have read and agreed to the published version of the manuscript.

**Funding:** This research received no external funding.

**Institutional Review Board Statement:** The study was conducted in accordance with the Declaration of Helsinki, and approved by the Institutional Review Board of University of Almería (protocol code UALBIO2022/035, 25 October 2022).

**Informed Consent Statement:** The study was conducted in accordance with the Declaration of Helsinki and approved by the Institutional Review Board of University of Almería (UALBIO2022/035).

**Data Availability Statement:** Data are available on request.

**Conflicts of Interest:** The authors declare no conflict of interest.

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
