# Peer review of "Relationships between Informal Sports Leadership and Emotional Intelligence: A Cross-Sectional Study"

_sustainability, doi:10.3390/su151914571_

Round 1

Reviewer 1 Report

The article is very interesting and touches on current issues related to the psychology of sport. The reviewer proposes to the authors a study using the same research tools in sports teams or representatives of various team disciplines. This may show that in different types of sports these issues are interpreted differently.

Author Response

REVIEWER 1

The article is very interesting and touches on topical topics related to the psychology of sport. The reviewer proposes to the authors a study using the same research tools on sports teams or representatives of different team disciplines. This may show that in different types of sports these issues are interpreted differently

Dear revisor, thank you very much for your contributions, which help substantially to improve this work. We are grateful for the advice and the time devoted to this work. All changes made appear in green.

We will have your contributions in consideration for future research in which we will employ as a sample of the research sports teams or representatives of various team disciplines. This has been added in the section on future lines of research (lines 614-627).

Reviewer 2 Report

Basic reporting

Dear authors, although the results of the study are interesting, I have some concerns that you need to address before acceptance and publication. 

An extensive spell, grammar and syntactic check seems appropriate to me, especially in the highlighted parts. Several paragraphs are difficult to understand.

Abstract

Missing conclusion

The results and method section should be improved with more specific statistical data.

keywords usually should be different from that used in the main title.

Introduction

I think the introduction is too extensive and should be better summarized.  The introduction must be concise, clear, and non-technical, the sentences and paragraphs must be short. Write for a general audience.  Detailed descriptions, speculations, and criticisms of particular studies should be included in the discussions.

The introduction must take the reader step by step from what is known to what is unknown. End with your specific question. Known → Unknown → Question/Hypothesis. Background, known information→ Knowledge gap→ Your question/aim/hypothesis, why your experimental approach is new and different and important.

The aim/s of the work is not clear, it should be summarized and stated in a more straightforward manner.

Minor:

When there are more than two citations you must report them in this way [1-3], otherwise, if there are just two citations you must report them in this way [1,2]. Please check the whole manuscript.

Line 50 and line 55: you need to change the citation style. Please check the whole manuscript.

Methods

The methods section is sufficiently described.

Table 1: I think you should use "course" in the first column, to make it clear what it refers to.

Validity of the findings

Table 2/3: You should add a note under the table, where you need to state the meaning of the abbreviation utilized, what asterisks refer to and what p-value was considered statistically significant.

The discussion section is quite limited, it should be expanded.  Studies presented in the introduction must be commented on, criticized, and compared with the results of the own study and with other previous studies.

Summarize the main results and then the secondary results in relation to the study purposes.

Explain what the data mean. State if the findings are novel. How does your study support or criticize current knowledge? Compare your results with other people’s results.

Discuss how your findings support or challenge the paradigm. Take-home message (Restate your main findings and the novelty of the paper according to the current literature to help better understand how this paper is different from others already published).

An extensive spell, grammar and syntactic check seems appropriate to me, especially in the highlighted parts. Several paragraphs are difficult to understand.

Author Response

REVIEWER 2

Dear authors, although the results of the study are interesting, I have some concerns that you should address before they are accepted and published. I think that a thorough spelling, grammatical and syntax review is appropriate, especially in the highlighted parts. Several paragraphs are difficult to understand.

  • thank you very much for your contributions which help substantially to improve this work. We are grateful for the advice and the time devoted to this work. All changes made appear in yellow.

Summary.

  • Missing conclusion.

It has been added that the conclusion of this research demonstrates the existence of a significant connection between the components of emotional intelligence (attention, clarity and regulation) and sports leadership (social support, empathy, sports values, decision-making and task orientation). Thus, this study concludes that this link is characterized by being direct and positive (lines 25-28)

  • The Results and Methods section should be enhanced with more specific statistical data.

It has been added in the summary that the main contribution of our research lies in the demonstration that there is a statistically significant relationship between clarity and emotional regulation with empathy (=0,18, p < 0,001).  Decision-making (=0,08, p < 0,001). Social support l(=0.19, p <0.001). And sports values (=0,01, p < 0,001). Clearly indicating the specific statistical data for each assessed dimension (lines 20-23)

  • Keywords should normally be different from those used in the main title.

The keywords have been modified and are now: leadership; sport; emotional skills; University students (lines 29)

Introduction

  • I think the introduction is too extensive and should be summarized better. The introduction should be concise, clear and non-technical, and the sentences and paragraphs should be short. He writes for a general audience. The introduction should take the reader step by step from what he knows to what he does not know.

In order to maintain the bibliographic solidity of our research, we have created a new section intended for the introduction, where the main ideas of our work are summarized (lines 34-56)

  • End with your specific question. Known → Unknown → Question/hypothesis. Background, known information → Knowledge gap → Your question/objective/hypothesis, why your experimental approach is new, different and important. The objectives of the work are not clear; they should be summarized and expressed in a simpler way.  Minor: When there are more than two quotes you should report them this way [1-3], otherwise, if there are only two quotations you must report them that way [1,2]. Please review the entire manuscript.

The general objective, research questions and hypotheses have been rewritten, indicating that the research question that this research directs is about knowing what the relationship between informal leadership and emotional intelligence is? So our main objective is to study the relationship between informal leadership and emotional intelligence.

As specific objectives we propose to analyse the influence of each of the dimensions of both psychological constructs.

As for social support, our goal is to analyze the relationship between social support and attention, emotional clarity and regulation and social support. And for this, the following hypothesis is formulated:

- There is a statistically significant relationship between attention, clarity and emotional regulation and social support.

As for empathy, our goal is to investigate the relationship between it and attention, emotional clarity and regulation, and social support. And for this, the following hypothesis is formulated:

- There is a statistically significant relationship between attention, clarity and emotional regulation and empathy.

As for values, our goal is to explore their influence on attention, clarity and emotional regulation and social support. And for this, the following hypothesis is formulated:

- There is a statistically significant relationship between attention, clarity and emotional regulation and sports values.

As far as decision-making is concerned, our goal is to study the relationship between it and attention, emotional clarity and regulation, and social support. And for this, the following hypothesis is formulated:

- There is a statistically significant association between attention, clarity and emotional regulation and decision-making.

And, as far as task orientation is concerned, our goal is to examine the relationship between it and attention, emotional clarity and regulation, and social support. And for this, it is formulated

Reviewer 3 Report

Dear Author, congratulations on your successful study. I recommend doing some minor revisions. This manuscript is suitable for publication after minor corrections.

1-Data collection tools should be specified in the abstract.

2- The sample size was calculated using Soper's models: What are the values used according to this model and how many sample size were found at least?

3- The text-to-text ratio of the discussion section is very low, only 1 page.

Author Response

REVIEWER 3

  • Dear author, congratulations on your successful study. I recommend making some minor revisions.
  • This manuscript is suitable for publication after minor corrections.

Dear revisor, thank you very much for your contributions, which help substantially to improve this work. We are grateful for the advice and the time devoted to this work. All changes are marked.

1-The data collection tools should be specified in the summary.

It has been added in the summary that two validated and standardized instruments were applied to measure each of the psychological constructs that form the thematic axis of this research. Thus, to measure self-perceived leadership was used the Sport Leadership Behavioral scale, and its Spanish version [42] Sports leadership behavior inventory [37]. On the other hand, the Trait Meta Mood Scale (TMMS-24) (lines 18-23) was used to measure the self-perceived emotional intelligence of participants.

2- The sample size was calculated using the Soper models: What are the values used according to this model and how many sample sizes were at least found?

Sample size was calculated using Soper's a priori sample size calculator for structural equation models (Soper, 2022). Thus, based on six observable variables, one latent variable, with an expected effect size of 0.30, a probability level of 0.05 and a desired statistical power level of 0.95, the recommended minimum effect size was 200 cases, i.e. the number of participants in our study was close enough to the suggested population size (lines 296-301)

3- The text-to-text ratio of the discussion section is very low, only 1 page.

The discussion has been rewritten to address all these issues

Reviewer 4 Report

Abstract

Line 15 Rephrase this sentence.

 Introduction

Lines 37-8. This needs to be referenced.

Lines 55-6. ‘‘…..model of leadership (Chelladurai, 55 1990; Mestre et al., 2018) the most prominent model for our study.’’ Authors should use the appropriate referencing.

Lines 62 -4. There are loads of repetition. Rephrase this.

Line 72. ‘As other works have already done [35-36-37-38].’ This is not a complete sentence.

Lines 70 -2 and Lines 101-3. It is not clear what the authors are focusing on. eg. Lines 70-2. 'However, in this study we are going to focus on informal leadership, which is theoretically based on the contributions of the role differentiation theory [30-31] and the role integration theory [32-33-34]'. Lines 101 -3. 'In this paper we focus 101 specifically on the skill models [56], which are characterized by the fact that they are com- 102 posed of different skills.........'

Lines 112- 51. This has been written as discussion. The authors are debating themselves.

Materials and Methods

Lines 171 - 174. This should be clearly explained. Rewrite this section.

Lines 186 - 187. Sample should not include results. Table 1 should be part of results.

Lines 216 - Authors should state clearly the 'validated and standardized instruments' or they can remove this paragraph.

Lines 247 -64. This is not necessary as it doesn't add anything to the paper.

Results

Generally, the authors over explained their results eg - lines 336 - 43 and 391-6. Some of their comments should be in discussion.

Discussion- More referencing is need in this section. 

Conclusion - This should a brief summary of all results and discussion. 

Generally, this manuscript need to be edited by a native English speaker. In some instances, too long sentences full of verbose. 

Author Response

REVIEWER 4

Dear reviewer, thank you very much for your contributions, which help substantially to improve this work. We are grateful for the advice and the time devoted to this work. All changes made appear in blue.

Abstract

  • Line 15 Reformulate this sentence.

The line was reformulated.

Introduction

  • Lines 37-8. It is necessary to make reference to this.

 Done

  • Lines 55-6. ".....leadership model (Chelladurai, 55 1990; Mestre et al., 2018) the most outstanding model for our study". Authors must use appropriate references.

This error has been corrected throughout the manuscript.

  • Lines 62-4. There are many repetitions. Rewrite this.

Reworded

  • Line 72. ‘As other works have already done [35-36-37-38]’. This is not a complete sentence.

It was a typo.

This typo was eliminated

  • Lines 101 -3. 'In this article we focus 101 specifically on skill models [56], which are characterized by the fact that they consist of 102 different skills.........'

Corrected

  • Lines 112 to 51. This has been written as a discussion. The authors discuss each other.

Removed

Materials and methods

  • Lines 171 - 174. This must be explained clearly. Rewrite this section.

reworded

  • Lines 186 - 187. The sample should not include results.

It's a mistake. Corrected

  • Table 1 should be part of the results.

Table 1 simply provides information on the main social and demographic characteristics of the sample. It does not yield any results from the study. It only clarifies the information of the sample characteristics.

  • Lines 216 - Authors must clearly indicate 'validated and standardized instruments' or may delete this paragraph.

deleted

  • Lines 247-64. This is not necessary as it adds nothing to the paper.

Dear reviewer, we appreciate your proposal, but the other four reviewers has indicated to us that this paragraph must exist and appear in the manuscript.

Results

  • In general, the authors explained their results too much, for example lines 336 - 43 and 391-6. Some of his comments should be discussed.

reworded

Discussion: More references are needed in this section.

Reworded

Conclusion: this should be a brief summary of all the results and discussion.

reworded

Reviewer 5 Report

Lines 170 - 175: This section does not start out the Methods section.  Move it after the Measures section, or add it to the procedures section.

The ‘sample’ section needs to be renamed “participants”

Line 189: There should a heading titled “procedures’.  Lines 247 - 265 are part of the procedures and needs to be consolidated with the paragraph that starts “First, a questionnaire….  Also you procedure section has Step 1… Step 2.  Write a paragraph. This doesn’t flow.  

Lines 207-212: Your Ethic statement, should just stated that you received IRB approval.  Everything you have in there is unnecessary.  

Your writing in English was good.

Author Response

REVIEWER 5

Dear revisor, thank you very much for your contributions, which help substantially to improve this work. We are grateful for the advice and the time devoted to this work. All changes made appear in pink.

  • Lines 170 - 175: this section does not begin with the Methods section. Move it after the Measures section or add it to the Procedures section.

Following your instructions, have been included in the procedure section (lines 391-393)

  • It is necessary to rename the "sample" section to "participants"

Modified

  • Line 189: There should be a heading entitled “procedures”.

Added

  • Lines 247 - 265 are part of the procedures and should be consolidated with the paragraph beginning with “First, a questionnaire.... Also your procedure section has Step 1... Step 2. Write a paragraph. This doesn't flow.

It has been modified.

  • Lines 207-212: Your statement of ethics should only indicate that you have received IRB approval. Everything you have there is unnecessary.

Reworded accordingly

Round 2

Reviewer 2 Report

The authors adressed all my previous comments. I have no further suggestion.

Author Response

Dear reviewer. 
We are very grateful for the time you spent reviewing our article, which helped immense in improving it

Reviewer 4 Report

Authors should rewrite the conclusion. They should remove the sentences which start with despite weakness..............' and summarises their findings only. 

Also, we encourage authors to include references in English only i.e., translate or include English version in the reference list.

Minor grammatical corrections are required.

Author Response

Dear reviewer.

We are very grateful for the time you spent reviewing our article, which helped immense in improving it. Please note that we have rewritten the conclusions according to your suggestions and translated the bibliographic references into English.

The amendments are written in red, and shaded in green

However, we warn that translations of bibliographic references may impair access to the articles, considering that the search engines may not recognize them.

Once again, thank you very much for your support in reviewing this article, which we now consider to be richer and more balanced.

Reviewer 5 Report

Thank you for addressed the suggested changes.  Your changes improved your manuscript.  Good work.

Author Response

Dear reviewer.

We are very grateful for the time you spent reviewing our article, which helped immense in improving it.